# Immunotherapy and Its Development for Gynecological (Ovarian, Endometrial and Cervical) Tumors: From Immune Checkpoint Inhibitors to Chimeric Antigen Receptor (CAR)-T Cell Therapy

**DOI:** 10.3390/cancers13040840

**Published:** 2021-02-17

**Authors:** Giuseppe Schepisi, Chiara Casadei, Ilaria Toma, Giulia Poti, Maria Laura Iaia, Alberto Farolfi, Vincenza Conteduca, Cristian Lolli, Giorgia Ravaglia, Nicole Brighi, Amelia Altavilla, Giovanni Martinelli, Ugo De Giorgi

**Affiliations:** 1Department of Medical Oncology, IRCCS Istituto Romagnolo per lo Studio dei Tumori (IRST) “Dino Amadori”, Via P. Maroncelli 40, 47014 Meldola, Italy; chiara.casadei@irst.emr.it (C.C.); marialaura.iaia@irst.emr.it (M.L.I.); alberto.farolfi@irst.emr.it (A.F.); vincenza.conteduca@irst.emr.it (V.C.); cristian.lolli@irst.emr.it (C.L.); nicole.brighi@irst.emr.it (N.B.); amelia.altavilla@irst.emr.it (A.A.); giovanni.martinelli@irst.emr.it (G.M.); ugo.degiorgi@irst.emr.it (U.D.G.); 2Clinical Oncology, Arcispedale Sant’Anna University Hospital, 44124 Ferrara, Italy; itoma.mf@ausl.fe.it; 3Istituto Dermopatico dell’Immacolata, IDI IRCCS, 00167 Rome, Italy; g.poti@idi.it; 4Medical Oncology Unit 1, University of Genoa, Ospedale Policlinico San Martino IRCCS, 16132 Genoa, Italy; 5Unit of Biostatistics and Clinical Trials, IRCCS Istituto Romagnolo per lo Studio dei Tumori (IRST) “Dino Amadori”, Via P. Maroncelli 40, 47014 Meldola, Italy; giorgia.ravaglia@irst.emr.it

**Keywords:** chimeric antigen receptor (CAR)-T cell therapy, lymphocytes, immunotherapy, ovarian cancer, endometrial cancer, cervical cancer

## Abstract

**Simple Summary:**

Gynecological cancers represent a group of malignancies with high incidence and mortality, despite their relative sensitivity to platinum-based chemotherapy. This review aims to illustrate the state of research in the field of immunotherapy, and in particular, deals with the development of CAR-T cell therapy, which represents a very promising treatment in the hematological field, but is still taking its first tentative steps in solid tumors.

**Abstract:**

Gynecological tumors are malignancies with both high morbidity and mortality. To date, only a few chemotherapeutic agents have shown efficacy against these cancer types (only ovarian cancer responds to several agents, especially platinum-based combinations). Within this context, the discovery of immune checkpoint inhibitors has led to numerous clinical studies being carried out that have also demonstrated their activity in these cancer types. More recently, following the development of chimeric antigen receptor (CAR)-T cell therapy in hematological malignancies, this strategy was also tested in solid tumors, including gynecological cancers. In this article, we focus on the molecular basis of gynecological tumors that makes them potential candidates for immunotherapy. We also provide an overview of the main immunotherapy studies divided by tumor type and report on CAR technology and the studies currently underway in the area of gynecological malignancies.

## 1. Introduction

Gynecological tumors have a high incidence. In particular, according to GLOBOCAN 2018 [1], cervical cancer (CC) is the fourth most commonly diagnosed tumor and the fourth cause of cancer death in females, worldwide, but ranks second in both incidence and mortality in lower income countries. With an estimated 295,414 cases and 184,799 deaths in 2018 worldwide, ovarian cancer (OC) represents 1.6% of all cancer incidence and 1.3% of all cancer deaths. With regard to endometrial cancer (EC), 382,069 new cases and 89,929 deaths were reported globally in 2018, representing 2.1% of all new cancer cases and 0.9% of all cancer deaths [1]. As is well known, gynecological cancers do not substantially benefit from classic chemotherapy regimens (with the exception, perhaps, of platinum-sensitive OCs) or from new target therapies. Therefore, the development of new therapies that can improve the survival of these patients is highly desirable.

The impressive results obtained with immunotherapy in other chemoresistant tumors such as melanoma and renal cancer have raised interest in the potential for using this approach in patients with gynecologic cancers. This has led to the development of a large number of clinical trials testing immunotherapy both as monotherapy and in combination with other strategies such as either chemotherapy, targeted agents, or both. To date, only microsatellite unstable (MSI-H) tumors and PD-L1-positive CCs have received Food and Drug Association (FDA) approval for treatment with immune checkpoint inhibitors.

Within this context, the development of chimeric antigen receptor (CAR) technology, which has led to exceptional results in the area of hematological cancers, represents a new opportunity for the treatment of gynecological cancers. However, the transfer of this technology to combat solid tumors has encountered some difficulties due to the intrinsic characteristics of these tumors. In this article we take an in-depth look at the molecular basis of gynecological tumors that make them potential candidates for immunotherapy. We also provide an overview of the main immunotherapy studies divided according to tumor type, and report on CAR technology and the studies currently underway in the area of gynecological cancers.

## 2. Immunotherapy in Gynecological Neoplasms: Rationale

### 2.1. Rationale for Immunotherapy in OC

The main reason for considering immunotherapy as a potential treatment against OC is because of to its higher expression of immune checkpoints (especially PD-L1) compared to other neoplasms, which is associated with poorer survival rate [2]. Abiko et al. demonstrated a close correlation between PD-L1 expression and peritoneal dissemination, suggesting a potential role of PD-L1-targeted immune checkpoint inhibitors in OC [3]. Moreover, tumor-infiltrating lymphocytes (TILs) are found in almost 50% of OCs. PD-L1 expression in tumors is believed to represent an anti-tumor immune response by the host: its correlation with clinical outcomes has been proven in several cancers with a high mutational burden [4,5,6]. This condition is frequently associated with BRCA1/2-mutated high-grade serous OCs, which harbor increased PD-1/PD-L1 expression with respect to homologous recombination-proficient tumors [7]. Such characteristics make OC a potential candidate on which to test CAR-T cell therapy.

### 2.2. Rationale for Immunotherapy in EC

In females, endometrial tissue shows specific correlations with the immune system. In fact, its main role is to protect the endometrium against sexually transmitted infections and other pathogens, but also to allow embryo implantation, and this process is finely regulated by sex hormone levels. The number of leukocytes varies considerably over the course of the menstrual cycle and their abundance before menstruation is probably due to their role in immunity defense during endometrial disruption. Conversely, the adaptive immune response is mediated by leukocyte aggregates (composed centrally by B cells and externally by T cells and macrophages), which are more frequent during the proliferative period, whereas they reduce their cytotoxic activities during the secretory period to allow conception [8].

The role of the tumor microenvironment has also been studied in EC, but its impact on prognosis is still unclear. However, a perivascular lymphocyte infiltrate would seem to be linked to poor prognosis rather than an intraepithelial infiltrate, which has been associated with survival [9]. Moreover, a number of other important correlations have been described: (1) correlation between high levels of CD8 in TILs and better overall survival (OS), and (2) correlation between CD8+ /FoxP3+ ratio and disease-free survival. Although T regulatory cells (Tregs) in TILs correlate with prognosis and myometrial infiltration, their presence alone has not been shown to influence OS [10,11,12]. Finally, EC represents a histology that often expresses the new immune checkpoints. In particular, Mo et al. reported that PD1 was expressed in TILs in 61% of EC cases and that PD-L1 expression was found in 80% of the same cells. The same authors found that almost 100% of metastatic EC expressed PD1 [13]. Thus, from a physiological point of view, the endometrium represents a good candidate to evaluate a treatment that is capable of modulating the immune response.

The role of immunotherapy in stimulating an individual’s immune response against cancer cells undoubtedly represents the future of cancer treatment. Several molecules directed against different biological targets have now been approved. Initially evaluated in melanoma patients, the impressive results obtained using immunotherapeutic agents has paved the way for their development in other tumor types [9]. The main problem lies in our scant knowledge of predictive biomarkers for immunotherapy. Within this context, there is evidence to suggest that the immune system can be activated in specific EC subtypes by the inhibition of the new immune checkpoints. For example, DNA polymerase epsilon (POLE)-mutated and mismatch repair (MMR)-deficient EC subtypes are thought to be likely to develop a significant immune response because of some of their characteristics. In fact, these subtypes harbor several DNA mutations that determine a high mutational burden which, in turn, results in a high antigen load that may also be responsible for the higher tumor lymphocyte infiltrate frequently seen in these tumors [14,15,16]. These two subgroups represent over 30% of ECs. Although MSI-H has also been identified in other gynecological malignancies, its frequency is fairly rare (3.5% of uterine carcinosarcomas, 2.6% of cervical squamous cell carcinomas and adenocarcinomas and 1.4% of high-grade serous ovarian cancer) [17,18,19]. A correlation between MSI-H and PD-1/PD-L1 expression in EC patients was reported by Howitt et al. [15] and could be exploited in clinical practice by enrolling patients with MSI-H pathway alterations in studies testing anti-PD1/PD-L1 molecules. Some interesting results have already been reported (Section 3.2).

In the light of what has been described thus far, EC, or at least some of its subgroups, could benefit from immunotherapy treatment and potentially also from CAR-T cell therapy. However, no studies have yet been conducted to date on the use of CAR-T in EC patients.

### 2.3. Rationale for Immunotherapy in CC

Theoretically, given that almost all CCs are human papillomavirus (HPV)-related tumors, CC could represent a paradigmatic example for the benefit obtained from immunotherapy. In fact, the immune system is often stimulated by non-human (viral) antigens, and for this reason it was possible to develop a vaccine as tumor prophylaxis [20,21]. Several studies have confirmed that a large number of genomic alterations are found in CC patients, for example, in the following genes: KRAS, PIK3CA, TP53 and PTEN. This high mutational burden may be responsive to immunotherapy. Moreover, HPV-integrated genes are often described in CCs [22]. Pan et al. identified 384 integrated gene sites that could influence T cell activation in the KEGG (Kyoto Encyclopedia of Genes and Genomes, https://www.genome.jp/kegg/ accessed on 17 February 2021) database, indicating the possibility of a strong correlation between HPV infection and immune surveillance [23].

There is an interesting correlation between HPV-mediated immune tolerance and tumor development. In particular, the ability of HPV to promote a so-called “non-lytic life cycle” unactivates (or partially activates) dendritic cell migration to lymph nodes and consequently inhibits immune activation. At the same time, low expression of E6 and E7 HPV proteins reduces Langerhans cell activity, leading to an immune-tolerant status that can potentially promote CC development [20].

With regard to immune checkpoints, high levels of CTLA4 and PD1/PD-L1 are often detected in CC patients, and PD1/PD-L1 are frequently expressed in dendritic cells in Cervical Intraepithelial Neoplasia (CIN) samples [24,25]. PD1/PD-L1 expression has been shown to be present in 95% of intraepithelial lesions and around 80% of squamous carcinomas [26,27,28]. Furthermore, several studies on CC have demonstrated high expression levels of immune-suppressive cytokines, such as IL-10, confirming an interesting link between immune checkpoints and CC progression [29]. Recently, D’Alessandris et al. showed that PD-L1 expression correlates with TILs, predicting response in CC patients treated with neoadjuvant chemotherapy [30].

## 3. Immunotherapy in Gynecological Neoplasms: Clinical Evidence

### 3.1. Immunotherapy in OC

Despite of the high prevalence of PD-L1 expression in OC, not all PD-L1-positive cases respond to treatment. In fact, response rates observed in trials testing immune checkpoint inhibitors as monotherapy were lower than expected (6–22%). Notwithstanding, prolonged responses are frequent [6]. The discrepancy between PD-L1 expression and response rate is probably a result of the high tumor mutational burden and strategies used by the tumor to escape immune surveillance, for example, the ability to create an immunosuppressive microenvironment [31]. Recently, several combination strategies have been tested to improve response rates and limit tumor-mediated immune suppression. The main strategies could be divided as follows: (1) chemo-immunotherapy combinations, (2) combination of immunotherapies (immune combo), (3) immunotherapy plus PARP-inhibitors and (4) antiangiogenic inhibitors plus immunotherapy (Table 1).

(1)Javelin 100 was a trial testing chemotherapy alone or in combination with anti-PD-L1 avelumab as first-line treatment. No differences between the two cohorts were reported in the interim analysis. The Javelin 200 trial tested the same combination in a second-line setting, reporting an objective response rate (ORR) of 9.6% and a median OS (mOS) of 11.2 months [6,32]. Wehnham et al. evaluated a second-line combination of weekly paclitaxel and anti-PD-1 pembrolizumab, observing an ORR of 51% and a 6-month progression-free survival (PFS) of 52% [33].(2)Zamarin et al. tested nivolumab alone or in combination with the anti-CTLA4 ipilimumab, reporting an overall response rate (ORR) of 12.2% (monotherapy cohort) vs. 31.4% (combo cohort), and an mOS of 21.8 months (monotherapy cohort) vs. 28.8 months (combo cohort) [34].(3)The TOPACIO-Keynote162 study investigated the combination of niraparib plus pembrolizumab, reporting an ORR of 25% [35]. The MEDIOLA trial, conducted on BRCA-mutated EC and testing the combination of olaparib plus anti-PD-L1 durvalumab, reported a disease control rate (DCR) of 81% [36]. The same combination was evaluated by Lee et al. in platinum-resistant OC, reporting a 37% DCR and an ORR of 14% [37].(4)Liu et al. tested the combination of nivolumab plus bevacizumab in recurrent EC, observing a median ORR (mORR) of 28.9% (ranging from 40% in platinum-sensitive EC to 16.7% in platinum-resistant disease) and a PFS of 8.1 months (ranging from 9.5 months in platinum-sensitive to 5.0 months in platinum-resistant disease) [38].

### 3.2. Immunotherapy in EC

The role of immunotherapy in EC has been demonstrated in MSI-H tumors. In 2015, Le et al. conducted a phase II clinical trial evaluating the anti-PD1 antibody pembrolizumab in patients with different tumors carrying alterations in the mismatch repair (MMR) gene pathway. The authors reported an ORR of 30–70% (even in the two patients with EC) [61]. Two years later, another trial conducted on fifteen EC patients by the same authors reported a DCR of 73% [39].

In 2016, Fader et al. tested pembrolizumab as monotherapy in patients with MMR-deficient EC, reporting a 56% ORR and an 88.9% DCR [40]. Based on these studies, pembrolizumab was approved by the Food and Drug Association (FDA) for use in the treatment of MMR-deficient solid tumors.

A phase II Japanese trial tested the anti-PD1 antibody nivolumab in 23 EC patients, demonstrating an ORR of 22.7%. The authors also conducted a molecular analysis, concluding that the ORR result was independent of the expression of PD-L1, and that all MSI-H cases obtained a response [41].

In 2019, the phase I/II GARNET trial tested dostarlimab (TSR-042), a new PD-1 inhibitor, in 94 patients with EC. The drug obtained an ORR of 27.7% and a DCR of 48.9%. The MSI-H subgroup was the most responsive (50% ORR), whereas the treatment proved effective in only 19.1% of cases with microsatellite-stable (MSS) disease. This new molecule was well tolerated, the most frequent side-effect being an increase in transaminase levels [42].

Fleming et al. evaluated the anti-PD-L1 antibody atezolizumab as monotherapy in a phase I study of fifteen EC patients, reporting an ORR of 13%, an mPFS of 1.7 months and an OS of 9.6 months [43].

Overall, these studies demonstrated the efficacy of immunotherapy in EC patients, even though numbers were small. Recently, several combination studies have been developed. Makker et al. carried out a phase Ib/II trial in which pembrolizumab was tested in combination with the tyrosine-kinase inhibitor lenvatinib in 53 EC patients. The authors reported an ORR of 39.5% and a 6-month DCR of 89%. The combination also induced a significant number of adverse events, in particular hypertension and diarrhea [44]. Notwithstanding, a phase III trial is currently underway to compare the same combination regimen with that of the investigator’s choice ((ClinicalTrials.gov accessed on 17 February 2021) Identifier: NCT03517449).

At the 2019 American Society of Clinical Oncology (ASCO) Annual Meeting, Rubinstein et al. reported the results of their phase II trial in which the anti-PD-L1 durvalumab alone or in combination with anti-CTLA4 tremelimumab was evaluated in a cohort of 56 patients with various EC histologies. Both strategies showed modest efficacy in this patient cohort. The cohort treated with durvalumab alone showed an ORR of 14.8% and a 6-month PFS of 13.3%, while the combination cohort achieved an ORR of 11.1% and a 6-month PFS of 18.5%. In the latter, grade 3–4 treatment-related adverse events were more frequent [45].

In addition to immune-checkpoint inhibitors, vaccines have also been developed for EC and tested in a number of clinical trials. The Wilms’ tumor gene, WT1, represents the main candidate for vaccines and has aroused great interest in the scientific community [62]. It was tested on two patients with human leukocyte antigen (HLA)-A∗2402-positive EC taking part in a phase II study but did not demonstrate efficacy [46]. Jackson et al. evaluated an HLA-A2-restricted, folate-binding protein (FBP)-derived peptide vaccine in 51 patients with different gynecological malignancies. The vaccine was administered at different dosages, but a benefit in terms of reduced risk of recurrence and 2-year DFS was only reached in those treated with the highest dose [47].

Another WT1-based vaccine was evaluated in three EC patients but did not obtain positive results [48].

Two other phase I studies tested different vaccinations, namely, the recombinant vaccinia-NY-ESO-1 (rV-NY-ESO-1) [49] and combination vaccines [50], in a small number of EC patients (one and two, respectively) but again, results were poor. Specifically, only one partial response was reported in the trial testing combination vaccines, while no responses were observed in the trial involving the rV-NY-ESO-1 vaccine.

The above is an overview of how immunotherapy could represent a therapeutic option for patients with EC, especially some subgroups. All the data discussed are summarized in Table 1.

### 3.3. Immunotherapy in CC

Numerous studies have assessed immune-checkpoint inhibitors in CC (Table 1). In a phase Ib KEYNOTE-028 trial, a cohort of 24 pretreated patients with CC-expressing PD-L1 received anti-PD-1 pembrolizumab, administered biweekly at a dosage of 10 mg/kg. An ORR of 17% and a DCR of 17% were reported in this cohort [51]. In the subsequent phase II KEYNOTE-158 trial, 98 CC patients (not only pretreated cases) were given pembrolizumab at a dosage of 200 mg every 21 days. The PD-L1 positive subgroup (83.3% of all cases) showed an ORR of 12% and a DCR of 30.6% [52]. Based on these results, in June 2018 the FDA approved pembrolizumab for use in patients with PD-L1-positive CC in progression after chemotherapy.

The anti-PD1 nivolumab has been evaluated in a number of CC studies, obtaining interesting results. In particular, one study assessed the antibody in two cases of neuroendocrine CC, a histology known to be associated with poor prognosis. Nivolumab induced a complete response in these patients, even though they both had PD-L1-negative disease. Similar promising results were subsequently obtained with a combination of nivolumab and stereotactic radiotherapy [63,64]. The phase I–II multicohort trial, CheckMate358, tested nivolumab at the biweekly dosage of 240 mg in 19 previously treated CC patients, reporting an ORR of 26% in this subgroup, regardless of PD-L1 expression [53]. However, another trial assessing nivolumab as subsequent-line therapy in previously treated CC patients reported poor results (only 4% ORR with 40% DCR) [54]. Similar results were obtained in a phase II trial testing the combination of anti-PD-L1 atezolizumab and anti-VEGF bevacizumab as subsequent-line therapy in 10 previously-treated CC patients (no responses and 50% DCR) [55].

The PD1/PD-L1 axis is not the only immune checkpoint that has been evaluated in CC. Some studies have focused on CTLA4 and its inhibitor, ipilimumab. The phase I GOG208 trial tested this antibody as post-chemo-radiotherapy in 19 patients with locally advanced CCs, reporting a 12-month disease-free survival (DFS) of 74% without significant adverse events [56]. Conversely, another phase I-II trial carried out on 34 previously treated CC patients only showed a 2% ORR and an OS of 8 months [57].

Immune-checkpoint inhibitors are currently under investigation in several ongoing studies, often as combination therapy [65].

Given that CC is an HPV-dependent tumor and that anti-virus prophylaxis has proven effective against CC development, several studies have also focused on vaccination therapy in metastatic disease.

Axalimogene filolisbac (ADXS11-001) is a live, attenuated *Listeria monocytogenes* (Lm) vaccine containing the HPV-16 E7 oncoprotein. It was tested in a phase II study of 109 patients with advanced CC, alone or in combination with a platinum-based chemotherapy regimen. The two cohorts showed similar results in terms of survival (17.1% and 14.7%, respectively) and treatment-related toxicities. Of note, the higher number of adverse events reported in the combination group were not attributable to the study treatment [58]. In another phase II trial, ADXS11-001 was administered as monotherapy in 50 patients with metastatic CC, obtaining a 2% ORR, 32% DCR and a 12-month OS of 38%. Almost all patients experienced treatment-related adverse events (TRAEs), which were grade 3–4 in 43% of cases [59]. In another trial, five patients with metastatic CC received the anti-PD1 pembrolizumab in combination with ADXS11–001, the authors reporting an ORR and DCR of 40%. Like the previous study, the majority of patients experienced TRAEs, grade 3–4 in 36% of cases [60].

## 4. CAR-T: Structure, Function and Toxicities

CAR-T cells are engineered T lymphocytes whose effector activity is powered by specific antibodies directed against specific antigens. For this reason, CAR-T cell activity does not depend on antigen presentation and so the engineered T cells bypass several pathways of immune tolerance [66]. CAR molecules consist of four fragments: (1) an extracellular domain (or ectodomain), which identifies specific tumor-associated antigens. This domain generally consists of a single-chain fragment variable (scFv) involved in specific antigen recognition mechanisms. It is composed of a variable (heavy and light) chain portion of an antibody and a linker (mainly composed of amino acid series such as glycine and serine, which confer flexibility and solubility to the structure, respectively) [67]. This section is joined to the (2) hinge domain, which is composed of amino acid fragments from CD8α, IgG1 and IgG4. Its role is to give flexibility and length to the CAR molecule, the latter important for CAR-T cell therapy efficacy [68]. For example, mesothelin CAR-T cells containing an IgG4 hinge domain have demonstrated higher efficacy and proliferation than hingeless CAR-T cells. Consequently, the hinge region brings mesothelin near the membrane, resulting in reduced steric inhibition between scFv and its target epitope [69]. The hinge domain is in turn joined to the T cell by a (3) transmembrane domain (consisting of a transmembrane domain of CD3, CD8, CD28 or FcεRI), in turn joined to the 4) intracellular domain, consisting of the intracytoplasmic domain of CD8, CD28 or CD137 and CD3ζ. The immune receptor tyrosine-based activation motif (ITAM) is located in this region and is involved in signal transduction mechanisms [70].

To date, five generations of CARs have been developed, each with distinct characteristics. In the first generation, CARs have only one receptor with the known tripartite division (ectodomain, transmembrane domain and intracellular zone). However, no significant results were reported for first generation CARs due to the lack of a costimulatory molecule [71,72]. Thus, in the second generation, a costimulatory molecule (CD28, CD27, CD134 or CDB7) was added to the first generation’s structure to prolong T-cell activation in circulating blood. For the same reason, a second costimulatory factor (CD28, 4-1BB, or CD3ζ) was inserted in the third CAR generation’s structure [73]. The fourth generation’s structure (also called TRUCK, in other words, T cells redirected for universal cytokine mediated killing) is different, the CARs consisting of both a costimulatory and proinflammatory molecule to improve T-cell efficacy [74]. The pro-inflammatory function can be performed by cytokines (such as IL-12, IL-15 or IL-18), knock-out genes (PD-1 or DGK) and knock-in genes (TRAC or CXCR4), controlled and inducible systems (Syn/Notch), and multiantigen combinations (HER-2 + IL13 Rα2). These molecules counteract the immunosilencing action (induced by the microenvironment) by inducing a shift in the immune response towards a T helper-1 type [75] and preventing antigen escape [76]. Recently, a fifth CAR generation was conceived: it is fairly similar to that of the fourth generation in terms of molecular structure (including CD3ζ, and CD28 as a costimulatory molecule, which enhances the secretion of proinflammatory cytokines), with the insertion of an intracellular domain of IL2RB chain fragment (or another similar receptor). This latter fragment was added to bind STAT3, which is responsible for cell proliferation, preventing terminal differentiation and ensuring CAR persistence [77]. Within this context, it is also possible to transplant these cells from healthy donors by knocking-out *HLA* and *TCR* genes to avoid graft-versus-host disease and to use the therapy for more than one patient [78].

The CAR-T strategy was initially conceived to search for a hypothetical receptor that could be used to re-target T cell responses against surface antigens [79]. Early data and clinical trials were performed targeting HIV viral antigen (CD4-CAR) and Carcinoembryonic Antigen (CEA) against tumors. These first constructs had several drawbacks, mainly the lack of persistence in vivo. However, in recent years, advances in the technique have led to the development of several CAR generations, which led to their first clinical successes in hematological cancers, culminating in the approval by the United States FDA of the first CAR-T cell therapies [80,81,82,83,84] (Table 2). Notwithstanding, CAR-T cell therapy has not shown the same results in solid tumors, despite the impressive results obtained with TIL-dependent immunotherapy. Several factors may be responsible for this [85], for example, immune-dependent cancer antigen selection, which potentially improves the proliferation of cancer cells not expressing molecular targets [86], reduced tumor trafficking [87], reduced CAR-T persistence in the patient [88], CAR-T destruction, which is dependent on the tumor microenvironment [89] and no evidence of cancer-specific antigens [90]. These conditions may lead to immune-related toxicity, including cytokine release syndrome, acute kidney injury, and tumor lysis syndrome, all of which can lead to severe nephropathy [91,92]. Other CAR-related toxicities have been reported in the literature, including anaphylaxis, graft-versus-host disease, neurological toxicity (including confusion, delirium, aphasia, myoclonus and seizures) and “on-target/off-tumor” toxicities [93].

To date, several studies are in the process of evaluating potential solutions against CAR-T toxicities: the main hypotheses being worked on are the following: (1) modification of the CAR molecule [94] (2) insertion of inducible suicide gene cassettes that lead to CAR-T cell apoptosis in the event of toxicity [95] (3) use of two CAR molecules with different functions (inhibitory versus stimulatory) or with divided sections (e.g., one molecule containing the TCR domain and the other containing the stimulatory domain) [96].

## 5. CAR-T Cell Therapy in Gynecological Tumors

Unlike hematological tumors in which CD19 represents an ideal antigen given its close correlation with B lymphocytes and its absence in other cell lines, there is no known antigen with similar characteristics for gynecological cancers. Mesothelin, CA125 and folate receptor are currently the most widely studied antigens in trials testing CAR-T cell therapy in these tumors (Figure 1).

### 5.1. CAR-T Cell Therapy in OC

The activity of anti-mesothelin CAR-T cell therapy was evaluated in a study on six relapsed OC patients (Table 3). The authors reported (1) the possibility of manufacturing CAR-T cells in all patients, (2) a transient expansion of CAR-T cells in the peripheral blood and (3) one-month stable disease [97]. In China, several studies are currently ongoing in OC patients; a phase I/phase II trial is testing an anti-mesothelin CAR molecule expressed by autologous T cells transduced by a retroviral vector in 20 patients with relapsed/refractory epithelial OC. The study end date is expected between 2022 and 2023 ((ClinicalTrials.gov accessed on 17 February 2021) Identifier: NCT03916679). Another phase I trial is evaluating the activity of LCAR-M23, a CAR-T cell therapy targeting mesothelin, in 34 patients with relapsed/refractory epithelial OC. Before LCAR-M23 infusion, these patients receive a premedication regimen (cyclophosphamide 300 mg/m^2^ and fludarabine 30 mg/m^2^ once daily for 3 d). Five to seven days after this regimen, only a single infusion of LCAR-M23 is scheduled. The results of this study are expected in 2024 ((ClinicalTrials.gov accessed on 17 February 2021) Identifier: NCT04562298).

A third phase I trial is currently ongoing at the Shanghai 6th People’s Hospital to test the safety and efficacy of anti-mesothelin CAR-T cell therapy in 10 patients with refractory/relapsed OC. All patients receive a premedication regimen comprising cyclophosphamide 300 mg/m^2^/d and fludarabine 30 mg/m^2^/d. They then receive anti-mesothelin CAR-T cells as a single injection at a dose of 5 × 106/kg on day 1. The results of this study are expected in 2022 ((ClinicalTrials.gov accessed on 17 February 2021) Identifier: NCT03814447).

Another phase I trial on OC is currently evaluating the safety and efficacy of CAR-T cells based on ten different tumor-specific antibodies in several tumor types; anti-mesothelin and anti-C-met CAR-T cells are among the treatments administered to patients. The results of this study are expected in 2023 ((ClinicalTrials.gov accessed on 17 February 2021) Identifier: NCT03638206).

An early phase I trial is currently evaluating the safety and efficacy of anti-mesothelin CAR-NK cells. Using a natural killer (NK) structure instead of a T-cell eliminates the need for costimulatory factors thanks to the Major Histocompatibility Complex (MHC)-independence of NK cells. This treatment will be administered to 30 patients with epithelial OC and study results are expected in 2021 ((ClinicalTrials.gov accessed on 17 February 2021) Identifier: NCT03692637 NCT03692637).

Another Chinese trial is underway to test an αPD1-mesothelin CAR-T cell therapy in 10 OC platinum-refractory patients. These are a particular type of anti-mesothelin CAR-T cells that secrete PD1 nanobodies. During CAR-T cell production, patients receive a premedication regimen (cyclophosphamide) to deplete lymphocytes. The estimated completion date of the study is scheduled for June 2022 ((ClinicalTrials.gov accessed on 17 February 2021) Identifier: NCT04503980).

A phase I trial is currently testing MCY-M11, an intraperitoneal anti-mesothelin CAR-T cell therapy in 27 patients with OC, fallopian tube or peritoneal carcinoma. Patients are divided into four cohorts that differ on the basis of the drug dosage administered and whether or not a premedication is infused. The results of the study are expected in 2022 ((ClinicalTrials.gov accessed on 17 February 2021) Identifier: NCT03608618).

In the U.S., a phase I study is ongoing to test the efficacy of PRGN-3005 UltraCAR-T, an autologous CAR-T cell therapy developed by Precigen, Inc. (Germantown, MD, USA). This treatment will be administered to 71 patients with platinum-resistant OC, fallopian tube and peritoneal carcinoma. The study results are expected in 2026 ((ClinicalTrials.gov accessed on 17 February 2021) Identifier: NCT03907527). A second phase I/II trial is underway to evaluating the safety and efficacy of anti-mesothelin CAR-T cell therapy in 15 OC patients, divided into five cohorts, each treated with a different dosage. The only response to date has been stable disease in a patient treated with a dosage of 3 × 10^6^ anti-mesothelin CAR-T cells + IL-2. Overall, therapy was well tolerated, with only a few grade ≥3 toxicities (one case of anemia, one case of lymphocytopenia, one case of thrombocytopenia, one case of hypoxia and one case of constipation) ((ClinicalTrials.gov accessed on 17 February 2021) Identifier: NCT01583686).

At the University of Pennsylvania, three trials are currently ongoing in OC patients; the first is a phase I study testing the safety and feasibility of intraperitoneally administered MOv19-BBz, a lentiviral transduced CAR-T cell therapy. The study is composed of 18 patients with OC, fallopian tube or peritoneal carcinoma divided into four cohorts that differ according to the drug dose administered and whether or not premedication is used. The estimated primary completion date is scheduled for October 2024 ((ClinicalTrials.gov accessed on 17 February 2021) Identifier: NCT03585764). The second trial, recently completed, was a nonrandomized, open label phase I dose-escalation trial testing the safety and efficacy of anti-mesothelin CAR-T cell therapy in 19 patients with OC, pleural mesothelioma or pancreatic carcinoma. Treatment was administered intravenously with or without a premedication regimen. Results have yet to be published ((ClinicalTrials.gov accessed on 17 February 2021) Identifier: NCT02159716). The third trial is fairly similar to the previous one; 18 patients with mesothelin-expressing cancers will be divided into six cohorts (now reduced to four following a permanent closure of two of the cohorts). The aim of this phase I study is to establish the safety and feasibility of lentiviral transduced fully human anti-mesothelin CAR-T (huCART-meso) cells with or without lymphodepletion. The estimated primary completion date is scheduled for March 2023 ((ClinicalTrials.gov accessed on 17 February 2021) Identifier: NCT03054298).

As previously stated, mesothelin is not the only target used in CAR-based trials of OC.

Two studies were activated to assess the role of HER-2 as a target for CAR-T cell therapy in various tumor types. One of them was withdrawn because of safety considerations ((ClinicalTrials.gov accessed on 17 February 2021) Identifier: NCT02713984), while the second is underway to enroll 15 patients with relapsed or refractory HER-2 positive solid tumors (including OC). The manufactured CAR-T cells (CCT303–406) will be administered intravenously in four different dosages (dose 1: 3 × 10^5^ CCT303–406/kg; dose 2: 1 × 10^6^ CCT303–406/kg; dose 3: 3 × 10^6^ CCT303–406/kg and dose 4: 1 × 10^7^ CCT303–406/kg). Results are expected in 2023 ((ClinicalTrials.gov accessed on 17 February 2021) Identifier: NCT04511871).

A non-randomized, open-label phase I/II trial is currently testing a CD70-binding CAR-T cell therapy in 124 patients with CD70-expressing tumors, including OC. All patients receive a non-myeloablative, lymphodepleting premedication regimen consisting of cyclophosphamide and fludarabine followed, on day 0, by anti-CD70 CAR-T cells and high-dose aldesleukin. An instrumental evaluation of treatment response will be performed 4–8 weeks after the infusion. Study results are expected in 2028 ((ClinicalTrials.gov accessed on 17 February 2021) Identifier: NCT02830724).

A recently activated (13 November 2020) trial is currently enrolling 20 OC and EC patients who will be treated with anti-placental alkaline phosphatase (ALPP) CAR-T cell therapy. Moderate to strong cytoplasmic and membranous staining of ALPP has been seen testicular and gynecological cancer, making it a potentially interesting cancer-specific antigen. Results are expected in 2023 ((ClinicalTrials.gov accessed on 17 February 2021) Identifier: NCT04627740).

Another phase I study is underway to test anti TnMUC1 CAR-T cells in tumors expressing TnMUC1 (including OC, fallopian tube and peritoneal carcinoma). The study is composed of a dose escalation phase for the purpose of identifying the optimal dose for CAR-T cell therapy, and an expansion phase in which the preliminary efficacy of this therapy will be assessed. The study is expected to recruit around 112 patients (40 in the first phase and 72 in the second phase) ((ClinicalTrials.gov accessed on 17 February 2021) Identifier: NCT04025216).

### 5.2. CAR-T Cell Therapy in EC

To date, apart from the aforementioned study—(ClinicalTrials.gov accessed on 17 February 2021) Identifier: NCT04627740—that is testing an anti-ALPP CAR-T cell therapy, there are no other ongoing trials evaluating CAR-T cell therapy against EC. However, there are several molecules that represent potential targets for biological treatments and that could also represent a target for CAR-T cell therapy. This is especially true for type II EC tumors, known for their poor response to current standard therapies and resultant poor prognosis. However, they are also the EC subgroup that has a higher percentage of mutations, and this feature could make them more suitable for CAR-T cell therapy if a specific target is found.

The ideal target should be as specific as possible to the tumor in question, which expresses it exclusively or in different quantities (amplification/overexpression) or qualities (mutations) with respect to normal tissue. One potential target for type II EC patients could be the Aα subunit of protein phosphatase type 2 A (PP2A). This subunit is encoded by PPP2R1A, a gene frequently mutated in up to 40% of patients with type II EC, whereas it is slightly more stable in type I EC patients, with mutations found in around 7% of cases. EC patients with mutated PPP2R1 A show a poor prognosis, mainly because of hyperactivation of the PI3K/Akt/mTOR pathway [98]. Moreover, PP2A can also influence the immune response through the negative regulation of the function of cytotoxic T-lymphocytes [99]. Thus, inhibiting PP2 A associated with CAR-T cell therapy could, in theory, stimulate the immune system against EC.

Human epidermal growth factor receptor 2 (HER-2) also represents a potential candidate target. Known for its overexpression in many breast and gastric cancers, HER-2 is also frequently overexpressed in EC and is very common in type II ECs [100] where its presence has been correlated with poor prognosis [101]. The first clinical trials testing trastuzumab, a monoclonal antibody targeting HER-2, in EC failed to obtain the same results that have seen in breast cancer, where it is currently used to treat HER2-positive patients. The reasons for this discrepancy in results between the two tumor types are not understood but are believed to be due to innate or rapidly acquired resistance to this anti-HER-2 monoclonal antibody [102]. Although this does not necessarily prevent HER-2 from being considered as a target for use in CAR-T cell therapy, it highlights the need for further research to understand the cause of this resistance before thinking about the selectivity of the CAR molecule towards HER-2. Several studies have hypothesized that the resistance is the result of overexpression of the PI3KCA pathway [103], which acts downstream of HER-2 [104]. In fact, PI3KCA gene mutations have been found in many aggressive ECs, especially those overexpressing HER-2. Furthermore, given that TDM1 has proven more active than trastuzumab in EC patients in terms of inhibiting mitosis and inducing G2/M phase cell cycle arrest through the action of DM1 toxin [105,106], associating CAR-T cell therapy with a similar molecule or with a PI3KCA inhibitor would probably help to overcome resistance against HER-2-targeting antibodies.

Another potential target is androgen receptor (AR), which is overexpressed in EC, especially in metastases. In fact, its expression differs substantially between primary and metastatic lesions. A specific EC subgroup with higher AR expression and lower estrogen receptor expression would seem to be correlated with poor prognosis [107] and could be used as a molecular target for these tumors. However, more research is needed to verify its potential role as a target for CAR-T cell therapy.

### 5.3. CAR-T Cell Therapy in CC

To date, three studies have focused on the use of CAR-T cell therapy in CC (Table 4). The first one tested anti-mesothelin CAR transduced peripheral blood lymphocytes in various tumor types expressing mesothelin, including CC. Patients were divided into five groups, each receiving a different treatment dosage. All patients received fludarabine 25 mg/m^2^/d intravenous for 5 days followed by CAR-T infusion (over 20 to 30 min, at different dosages: arm 01 = 1 × 10^6^ + IL-2, arm 02 = 3 × 10^6^ + IL-2, arm 03 = 1 × 10^7^ + IL-2, arm 04 = 3 × 10^7^ + IL-2 and arm 05 = 1 × 10^8^ + IL-2), intravenous cyclophosphamide 60 mg/kg/d for 2 days in 250 mL of 5% dextrose in water with a one-hour infusion of mesna 15 mg/kg/d for 2 days and intravenous aldesleukin 72,000 IU/kg three times a day starting within 24 h of CAR-T administration and continuing for up to 5 days. Of the 15 patients enrolled, only one stable disease was observed (arm 02). TRAEs were mild, including anemia, thrombocytopenia, fatigue, sinus arrhythmia and hypotension. This study terminated due to slow accrual ((ClinicalTrials.gov accessed on 17 February 2021) Identifier: NCT01583686).

An open-label, single-arm, prospective study is currently ongoing to test anti-PD-L1 armored anti-CD22 CAR-T/CAR-TILs targeting patients with solid tumors, including CCs. Therapy consists of an anti-CD22 CAR structure carrying an scFv of anti-PD-L1 monoclonal antibody. These cells are subsequently transplanted back into patients, the CAR molecule targeting CD22 + B cells in the blood. This results in the activation and expansion of CAR-positive cells, which secrete anti-PD-L1 scFv outside cells to stimulate anti-tumor activity. The final data collection is scheduled for August 2023 ((ClinicalTrials.gov accessed on 17 February 2021) Identifier: NCT04556669).

More recently, a third trial was activated to collect peripheral blood mononuclear cells of patients with GD2-, PSMA-, Muc1- or mesothelin-positive CCs. These cells will be selected through apheresis, activated and modified to CC-specific CAR-T cells. The estimated study completion date is December 2020 ((ClinicalTrials.gov accessed on 17 February 2021) Identifier: NCT03356795).

## 6. Discussion

The results of the use of CAR technology in gynecological cancers are disappointing. This is probably due to various factors, including a still relatively poor understanding of the method, especially in this type of tumor, patient selection and choice of the appropriate antigen. In fact, unlike hematological tumors, the antigens expressed by solid tumors are often non-specific, which is the basis of the “on-target/off-tumor” toxicity. Thus, given the lack of specific antigens, suicide genes have been inserted into some types of CAR molecules, in order to induce selective apoptosis of CAR-T cells and avoid the risk of unmanageable toxicities [108].

The poor results may also be related to the rate of other toxicities, namely, prolonged lymphopenia, infections, cytokine release syndrome (CRS) and neurotoxicity, all of which are frequent with the CAR-T cell treatment. Their management represents an important challenge to the successful development of this therapy.

Yet another concern is linked to the action of CAR-T in solid tumors. Unlike hematological malignancies, solid tumors are located in specific sites (not always near blood vessels) and are frequently protected by an immune-silencing microenvironment. Thus, CAR-T cell therapy may be negatively influenced by the tumor microenvironment, potentially culminating in CAR-T cell exhaustion. Specific measures are currently being investigated to counteract this inhibition and increase antitumor efficacy. For example, some CAR molecules are aided by the co-expression of cytokines such as IL-15, which play a significant role in T cell persistence and proliferation [109]. It has also been seen that the Nr4 a family of transcription factors are capable of modulating CAR exhaustion. In particular, preclinical studies in mice have shown that CAR-T cells not expressing these molecules are able to avoid exhaustion and bring about a reduction in tumor volume [110]. A dual-receptor CAR has also been conceived to improve CAR persistence in solid tumors, enabling CAR-T cells to express two artificial receptors for both tumor antigen recognition and growth stimulation.

The high variability in the immune landscape probably represents the most important problem limiting the use of CAR-T cell therapy in EC. In fact, heterogeneous antigen expression in different tumor sites is often observed [86]. With this in mind, new-generation CAR-T cells also have been modified in other ways, for example, by the transduction of a PD-1-targeting scFV and by the expression of interleukin-12 and other cytokines involved in immune-stimulation mechanisms [6].

As mentioned previously, changes in the structure of the CAR molecule also form the basis of toxicity-containment strategies. This, together with the known difficulties caused by the hostile action of the tumor microenvironment, represent important challenges along the road to the approval of the clinical use of CAR-T therapy in gynecological tumors.

## 7. Conclusions

The development of CAR technology in gynecological cancers is still in its early stages. As also seen for other cancer types, the road to using this therapy in solid tumors is much more complex than in hematological malignancies and must overcome various obstacles, the most important being the negative influence of the tumor microenvironment. It is hoped that, in the near future, the development of strategies to manage these problems will finally lead to results comparable to those obtained in the area of hematology.

## Figures and Tables

**Figure 1 cancers-13-00840-f001:**
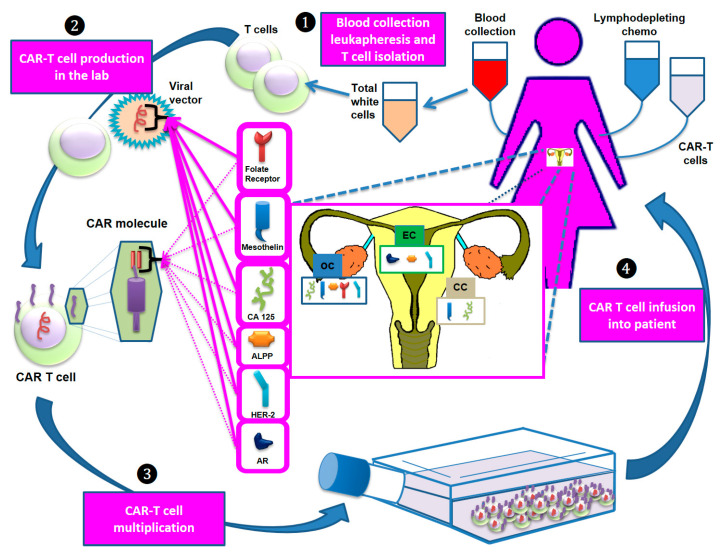
The main steps of the development of CAR-T cell therapy. Step 1: T cells are collected via apheresis. Step 2: The collected T cells are sent to a laboratory and genetically engineered by introducing specific DNA genes through a viral vector to produce CARs on the T cell surface. These molecules are designed to bind specific antigens (e.g., mesothelin, CA125) that recognize and destroy the tumor cells expressing them. Step 3: The re-engineered CAR-T cells are multiplied, frozen and (Step 4) reinfused into the patient. Before reinfusion, lymphodepleting chemotherapy is administered. Abbreviations: ALPP: placental alkaline phosphatase; AR: androgen receptor; CC: cervical cancer; EC: endometrial cancer; HER-2: human epidermal growth factor receptor 2; OC: ovarian cancer.

**Table 1 cancers-13-00840-t001:** Reported immune checkpoint inhibitors and vaccine therapy studies in gynecological cancer.

Study	Phase	Treatment	Number of Patients	Patient Population	Results	TRAEs (%)
OC						
Chemo-IMT combo						
JAVELIN Ovarian 200 [32]	3	Arm 1: Avelumab alone Arm 2: PLD alone Arm 3: Avelumab + PLD	566	Platinum- resistant/ refractory OC	ORR 3.7% vs. 4.2% vs. 3.3% PFS 1.9 vs. 3.5 vs. 3.7 mos OS 11.8 vs. 13 vs. 15.7 mos	PPE syndrome (9.9) Rash (9.3) Neutropenia (9.3) Fatigue (7.1) Stomatitis (5.5)
Wenham et al. [33]	2	Weekly paclitaxel (80 mmg/m^2^) + pembrolizumab 200 mg q3 w	37	Recurrent platinum- resistant EO, more than 3 prior therapies	ORR 51.4% DCR 86.50% mPFS 7.6 mos mOS 13.4 mos	Neutropenia Nausea/vomiting Edema, diarrhea Dyspnea Neuropathy Abdominal pain
IMT combo						
Zamarin et al. [34]	2	ARM 1 Nivolumab alone ARM 2 Nivolumab + Ipilimumab	49 51	Recurrent OC	ORR 31.4%, 28.1 mos 12.2%, 21 mos	Colitis/Diarrhea (16% vs. 4%) Anemia (16% vs. 4%) Rash (14% vs. 4%)
IMT-PARPi combo						
TOPACIO [35]	1/2	Pembrolizumab 200 mg q3 w + Niraparib 200 mg q day	60	Recurrent OC	ORR 18% DCR 65% mPFS 3.4 mos, (6-mo 31% and 12-mo 12%).	Fatigue Anemia Nausea Constipation Myelosuppression
MEDIOLA [36]	2	Olaparib 300 mg BID × 4 w, then 300 mg BID + Durvalumab 1.5 g q4 w	32	gBRCAm platinum- sensitive relapsed OC	ORR 63% DCR 81%	Anemia, Neutropenia, Increased amylase/lipase
Lee et al. [37]	2	Durvalumab 1500 mg q4 w + Olaparib 300 mg BID	35	Platinum- resistant/ refractory OC	ORR 14.7% DCR 52.9%	Anemia, Lymphopenia Atrial fibrillation Nausea
IMT-VEGFi combo						
Liu et al. [38]	2	Bevacizumab 10 mg/kg + Nivolumab 240 mg q 2 w until PD	38	Platinum sensitive/ resistant OC	ORR 28.9% DCR 34.2% mPFS 8.1 mos	Fatigue AST/ALT elevation Myalgia
EC						
IMT						
Le et al. [39]	2	Pembrolizumab 10 mg/kg q2 w	15	MMR- deficient EC with PD	ORR 53% DCR 73%	Colitis/Diarrhea Pancreatitis Hyperamylasemia
Fader et al. [40]	2	Pembrolizumab 10 mg/kg q2 w	9	Recurrent/ persistent MMR- deficient EC	ORR 56% DCR 88.9%, 1 y OS 89%	No grade 3
Katsumata et al. [41]	2	Nivolumab 240 mg q2 w	23	Advanced/ recurrent EC	ORR: 23% PFS 3.4 mos 1 y OS 48.5%	Pruritus Increased lipase Diarrhea
GARNET [42]	1/2	Dostarlimab 500 mg q3 w × 4, then 1000 mg q6 w	94	Recurrent/ persistent EC	ORR 27% (50% MSI-H 19.1% MSS) DCR 48.90%	AST elevation
Fleming et al. [43]	1 A	Atezolizumab 1200 mg q3 w	15	Advanced/ recurrent EC	ORR 13% DCR 26% mPFS 1.7 mos mOS 9.6 mos	Favorable safety profile
Makker et al. [44]	2	Pembrolizumab 200 mg q3 w + Lenvatinib 200 mg/die	53	Stage IV EC	ORR 39.6% DCR 86.8% PFS 7.4 mos	Hypertension Diarrhea Fatigue Hypothyroidism
Rubinstein et al. [45]	2	ARM 1 Durvalumab 1500 mg q4 w ARM 2 Durvalumab 1500 mg q4 w + Tremelimumab 75 mg q4 w	56	Recurrent/ persistent EC	ORR 14.8%, 6 mos PFS, 13.3 mos 11.1%, 18.5 mos	Fatigue Diarrhea Nausea/vomiting Pruritis
Vaccines						
Ohno et al. [46]	2	HLA-A^∗^2402- restricted adjuvant WT1 peptide 3 mg 1/w × 12 w	2	HLA-A^∗^2402-positive EC resistant to standard therapy	ORR 0%, DCR 0%	Grade 1-2 erythema at injection site
Jackson et al. [47]	1/2 A	HLA-A2 restricted FBP-derived peptide at different dosage: 100 mcg/0.5 mL 500 mcg/0.5 mL 1000 mcg/0.5 mL + 250 mcg/1.0 mL GM-CSF	51	HLA-2+ Gyn. cancers	2-y DFS 43% for 1000 mcg dosage	Erythema at injection site Pruritus
Coosemans et al. [48]	1/2	Autologous DC electroporated with WT1 mRNA	3	Stage IV EC	ORR 0%, DCR 0%	Grade 1–2 erythema at injection site
Jager et al. [49]	1	rV-NY-ESO-1 3.1 × 10^7^ PFU × 2, then rV-NY-ESO-1 7.41 × 10^7^ PFU q4 w	1	Stage IV NY-ESO tumors	ORR = 0%, DCR = 0%	Grade 1-2 erythema at injection site
Kaumaya et al. [50]	1	Two HER2 B-cell epitopes combined with a T-cell epitope with n-MDP adjuvant 0.25 or 0.5 mg q3 w × 3	2	Stage IV EC	ORR = 50% (1 PR)	Diarrhea Hyperglycemia
CC						
IMT						
KEYNOTE-028 [51]	1 B	Pembrolizumab 10 mg/kg q2 w × 2 y	24	PD-L1+ previously treated CC	ORR 17% DCR 17% mPFS 2 mos mOS 11 mos	Rash, Fever, Proteinuria
KEYNOTE-158 [52]	2	Pembrolizumab 200 mg q2 w × 2 y	98	Pretreated CC patients	ORR 12.2% DCR 30.6% mPFS 2.1 mos mOS 9.4 mos	Hypothyroidism, Hyporexia, Fatigue
CheckMate 358 [53]	1/2	Nivolumab 240 mg q2 w	19	Pretreated CC patients	ORR 26.3% DCR 70.8% mPFS 5.5 mos	Diarrhea, Fatigue, Pneumonitis, Stomatitis Abdominal pain
Santin et al. [54]	2	Nivolumab 3 mg/kg q2 w	25	Persistent or recurrent CC	ORR 4% DCR 40%	Hepatotoxicity Type-1 diabetes
Friedman et al. [55]	2	Atezolizumab 1200 mg q3 w + Bevacizumab 15 mg/kg q3 w	10	Stage IV CC	DCR 50% mPFS 2.9 mos mOS 9 mos	Arachnoiditis, Hypoacusia, Weakness, Thrombosis
GOG208 [56]	1	Weekly Cisplatin (40 mg/m^2^) + extended field radiation, then Ipilimumab 3 mg/kg, 10 mg/kg, expansion cohort of 10 mg/kg	19	Stage IB2-IVA (n+) CC undergoing chemo- radiation	DCR 74%	Hyperlipasemia, Neutropenia, Rash
Lheureux et al. [57]	1/2	Ipilimumab 3 mg/kg q3 w × 4 cycles or 10 mg/kg q3 w × 4 cycles followed by maintenance q12 w)	42	Stage IV CC	ORR 2.9% DCR 32.4% mPFS 2.5 mos mOS 8.5 mos	Diarrhea
Vaccines						
Basu et al. [58]	2	ADXS11–001 (3 times) 1 × 10^9^ CFUs (80 mL infusion on day 1, 29, 57) Combo therapy ADX-011 (day 1) + weekly Cisplatin (40 mg/m^2^) post-vaccine 4 w × 5 w, then 1 cycle of ADS11–011 (3 times)	69	Stage IV CC	ORR: 17.1% mPFS 6.2 mos mOS 8.5 mos	Fever
Huh et al. [59]	2	ADXS11–001 (1 × 10^9^ CFU) q3 w × 3 doses (stage 1) or 1 y (stage 2)	50	Stage IV CC	ORR 2% (1 CR) 1 y OS 38%	Fatigue Nausea, Anemia
Cohen et al. [60]	1/2	ARM 1 ADXS11–001 ARM 2 Pembrolizumab ARM 3 combo	5	Stage IV HPV+ CC	ORR 40% DCR 40%	Chills, Fever, Nausea, Hypotension, Diarrhea, Fatigue, Tachycardia, Headache

Abbreviations: ADXS11–001: axalimogene filolisbac; ALT: alanine aminotransferase; AST: aspartate aminotransferase; DC: dendritic cell; DCR: disease control rate; FBP: folate-binding protein; gBRCAm: germline BRCA1 and/or BRCA2 mutation; Gyn: gynecological; HLA: human leukocyte antigen; IMT: immunotherapy; MMR: mismatch repair; mos: months; *n*: number of patients; n-MDP: nor-muramyl-dipeptide; ORR: overall response rate; (m)OS: (median) overall survival; PARPi: poly(ADP-ribose) polymerase inhibitors; PD: progressive disease; (m)PFS: (median) progression-free survival; PFU: plaque-forming unit; PLD: pegylated liposomal doxorubicin; PPE: palmo-plantar erythrodysesthesia; rV-NY-ESO-1: recombinant vaccinia-NY-ESO-1; TRAEs: treatment-related adverse events; VEGFi: vascular endothelial growth factor inhibitors; WT1: Wilms’ tumor-1; w: weeks; y: year.

**Table 2 cancers-13-00840-t002:** Current FDA-approved CAR-T cell therapies.

Drug [Reference]	Trade Name	Results	Approved For	Patient Population	Date of Approval
Tisagenlecleucel [80,81]	Kymriah	Complete Remission 90%	B-cell precursor acute lymphoblastic leukemia	Up to 25 years	30 August 2017
		Complete Remission 38% ORR 52%	Large B-cell lymphoma	Adult	1 May 2018
Axicabtagene ciloleucel [82]	Yescarta	Complete Remission 51% ORR 82%	Large B-cell lymphoma	Adult	18 October 2017
Brexucabtagene autoleucel [83]	Tecartus	ORR 87% (Complete Response 62%)	Relapsed or refractory mantle cell lymphoma	Adult	24 July 2020

**Table 3 cancers-13-00840-t003:** Ongoing clinical trials testing CAR-T cell Therapy in OC.

Study	Phase	Treatment	*N*	Patient Population	Endpoints	Locations	Current Status
MESO							
NCT03916679	1/2	anti-MESO CAR-T cells	20	MESO-positive OC patients	(1) Safety (2) ORR, PFS	Zhejiang University, Zhejiang, China	Recruiting
NCT04562298	1	LCAR-M23, anti-MESO CAR-T	34	Relapsed and Refractory Epithelial OC	Dose-limiting toxicity and TRAEs	Shanghai East Hospital Shanghai, China	Recruiting
NCT03692637	1	anti-MESO CAR-NK Cells	30	MESO-positive patients with stage II-IV epithelial OC	Occurrence of TRAEs	Allife Medical Science & Technology Co., Ltd.	Not yet recruiting
NCT04627740	1/2	Retroviral vector- transduced autologous T cells to express anti-ALPP CARs	20	ALPP-Positive Metastatic OC and EC	(1) Safety (2) ORR, PFS	Xinqiao Hospital of Chongqing Chongqing, China	Not yet recruiting
NCT04503980	1	MESO CAR-T Cells Secreting PD-1 Nanobodies	10	MESO-positive advanced solid tumors	(1) Dose-limiting toxicity (2) ORR, PFS, OS MTD	Shanghai Tenth people’s Hospital, Shanghai, China	Recruiting
NCT03814447	1	anti- MESO CAR-T cells	10	Refractory/Relapsed OC	(1) TRAEs (2) ORR, PFS	Shanghai 6 th People’s Hospital, Shanghai, China	Recruiting
NCT03608618	1	MCY-M11	27	Platinum resistant high grade serous OC, Fallopian and peritoneal carcinoma	(1) TRAEs (2) RECIST and irRECIST	Multiple Institutions in the USA	Recruiting
NCT02159716	1	anti-MESO CAR-T cells	19	MESO-positive advanced solid tumors	Occurrence of TRAEs	University of Pennsylvania, Philadelphia, USA	Completed
NCT03054298	1	huCART-MESO cells +/− CTX and different administrations	18	MESO-positive advanced solid tumors	(1) TRAEs (2) RECIST PFS, OS	University of Pennsylvania, Philadelphia, USA	Recruiting
NCT01583686	1/2	anti-MESO CAR-T cells	15	MESO-positive advanced solid tumors	ORR and TRAEs	National Cancer Institute, Bethesda, USA	Terminated due to slow/ insufficient accrual.
MUCINS (CA125)							
NCT03907527	1	PRGN-3005 UltraCAR-T cells	71	Advanced Stage Platinum Resistant OC	TRAE Incidence MTD	Fred Hutchinson Cancer Research Center, USA	Recruiting
NCT04025216	1	TnMUC1- Targeted Genetically- modified CAR- T Cells	112	Advanced TnMUC1-Positive Solid Tumors and Multiple Myeloma	(1) Dose Identification and ORR (2) Safety and tolerability	Multiple Institutions in the USA	Recruiting
OTHER TARGETS							
NCT04511871	1	CCT303–406 CAR modified autologous T cells	15	Relapsed or refractory stage IV metastatic HER2-positive solid tumors	(1) MTD (2) ORR, DCR, DOR, PFS TRAEs	Fudan University, Shanghai, China	Recruiting
NCT03585764	1	MOv19-BBz CAR-T cells	18	Alpha Folate Receptor -positive OC, Fallopian and peritoneal carcinoma	(1) TRAEs (2) ORR, PFS, OS	University of Pennsylvania, Philadelphia, USA	Recruiting
NCT02830724	1/2	Anti-hCD70 CAR-T cells	124	CD70 Expressing cancers	(1) TRAEs (2) ORR	National Cancer Institute, Bethesda, USA	Recruiting
NCT03638206	1/2	Multi-target Gene-modified CAR-T/TCR-T cell for Malignancies	73	Multi-target Gene-modified CAR-T/TCR-T Cell for Malignancies	(1) TRAEs (2) Clinical response	Zhengzhou University, Zhengzhou, China	Recruiting

Abbreviations: ALPP: placental alkaline phosphatase; DCR: disease control rate; DOR: duration of response; MTD: maximum tolerated dose; MESO: mesothelin; ORR: overall response rate; PFS: progression-free survival; Ph: Phase; (ir)RECIST: (immune-related) Response Evaluation Criteria in Solid Tumors; TRAEs: treatment-related adverse events.

**Table 4 cancers-13-00840-t004:** Ongoing clinical trials testing CAR-T cell therapy in CC.

Study	Phase	Treatment	No.	Patient Population	Endpoints	Locations
NCT01583686	1/2	anti-MESO CAR-T cells	15	Mesothelin-positive CC patients	Safety	National Cancer Institute, Bethesda, USA
NCT04556669	1	Autologous aPD-L1 armored anti-CD22 CAR-T cells	30	Refractory CC and other solid tumors	(1) Safety (2) ORR, PFS	4 th Hospital of Hebei Medical University, Shijiazhuang, China
NCT03356795	1/2	CC-specific CAR-T cells	20	GD2, PSMA, Muc1, Mesothelin or other markers positive CC	(1) Safety (2) ORR	Shenzhen Geno-immune Medical Institute, Shenzhen, China

Abbreviations: MESO: mesothelin; Muc-1: mucin 1; ORR: overall response rate; PFS: progression-free survival; PSMA: prostate-specific membrane antigen.

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
