# Peer review of "Immunotherapy and Its Development for Gynecological (Ovarian, Endometrial and Cervical) Tumors: From Immune Checkpoint Inhibitors to Chimeric Antigen Receptor (CAR)-T Cell Therapy"

_cancers, 2021, doi:10.3390/cancers13040840_

Round 1

Reviewer 1 Report

The authors answered all my comments. 

Author Response

The authors answered all my comments.

Reply: We very much appreciate the reviewer's reply.

Reviewer 2 Report

The manuscript is much improved now. Inclusion of tables provides a much clearer overview of published and ongoing clinical trials for immunotherapy and CAR-T in gynecological therapies. 

There are still some minor issues that should be addressed:

(1) There are still grammatical errors throughout the manuscript that need to be remedied. There are for example some very long run on sentences that use far too many colons and semi-colons; the semi-colon should be used judiciously. (eg line 79, 158, 299-309) 

(2) Line 295 - "CAR-T cells represent are engineered..."

(3) Line 313 - Paragraph should be broken somewhere. Perhaps before talking about generations of CARs

(4) Line 343 - Sentence should specifically refer to the "United States FDA"

(5) Table 2 - CR rates as should should be referenced

(6) Figure 1 - I like the new figure, it is very helpful for the review but it looks like a rough sketch for a figure rather than a final publication quality figure. As this is central to the review, a higher quality version of the figure must be made. Perhaps information on the tissue of origin for OC, EC, and CC could also be included. Additional targets discussed in the review (ALPP, HER2) could also be shown in the diagram. 

(7) New tables are very helpful. The formatting is a bit inconsistent, but I assume that would be fixed in the proof stage. 

(8) Line 543 - Discussion of toxicity and the use of suicide genes is confusingly worded. This should be re-written to clearly explain why a suicide strategy addresses on-target off-tumor toxicity. 

(9) Line 551 - Solid tumors are locating in specific tissue sites sides, and may or may not be well vascularized. 

Author Response

The manuscript is much improved now. Inclusion of tables provides a much clearer overview of published and ongoing clinical trials for immunotherapy and CAR-T in gynecological therapies. There are still some minor issues that should be addressed:

(1) There are still grammatical errors throughout the manuscript that need to be remedied. There are for example some very long run on sentences that use far too many colons and semi-colons; the semi-colon should be used judiciously. (eg line 79, 158, 299-309)

Reply: ok, we have modified as follows:

  • lines 79-80: “PD-L1 expression in tumors is believed to represent an anti-tumor immune response by the host: its correlation with clinical outcome has been proven in several cancers with a high mutational burden”
  • lines 161-165. We have removed semicolons as follows: “Recently, several combination strategies have been tested to improve response rates and limit tumor-mediated immune suppression. The main strategies could be divided as follows: 1) chemo-immunotherapy combinations 2) combination of immunotherapies (immune combo) 3) immunotherapy plus PARP-inhibitors and 4) antiangiogenic inhibitors plus immunotherapy (Table 1).”
  • Lines 302-305. We have removed colons as follows: “The hinge domain is in turn joined to the T cell by a 3) transmembrane domain (consisting of a transmembrane domain of CD3, CD8, CD28 or FcεRI), in turn joined to the 4) intracellular domain, consisting of the intracytoplasmic domain of CD8, CD28 or CD137 and CD3ζ.”
  • Lines 337-340. We have removed semicolons as follows: “Several factors may be responsible for this, e.g. immune-dependent cancer antigen selection, which potentially improves the proliferation of cancer cells not expressing molecular targets, reduced tumor trafficking, reduced CAR-T persistence in the patient, CAR-T destruction, which is dependent on the tumor microenvironment and no evidence of cancer-specific antigens.”
  • lines 504-508. We have modified as follows: “All patients received fludarabine 25 mg/m2/day intravenous for 5 days followed by CAR-T infusion (over 20 to 30 minutes, at different dosages: Arm 01 = 1x106 + IL-2, Arm 02 = 3x106 + IL-2, Arm 03 = 1x107 + IL-2, Arm 04 = 3x107 + IL-2, Arm 05 = 1x108 + IL-2), intravenous cyclophosphamide 60 mg/kg/day for 2 days in 250 mL of 5% dextrose in water with a one-hour infusion of mesna 15 mg/kg/day for 2 days and intravenous aldesleukin 72,000 IU/kg three times a day starting within 24 hours of CAR-T administration and continuing for up to 5 days”.

(2) Line 295 - "CAR-T cells represent are engineered..."

Reply: in the latest version of our manuscript, we have modified that sentence as follows (line 289): “CAR-T cells are engineered …”

(3) Line 313 - Paragraph should be broken somewhere. Perhaps before talking about generations of CARs

Reply, we have broken that paragraph between lines 306 and 307, before talking about generations of CARs

(4) Line 343 - Sentence should specifically refer to the "United States FDA"

Reply: ok, we have modified as follows (lines 334-335): “…culminating in the approval by the United States FDA of the first CAR-T cell therapies…”

(5) Table 2 - CR rates as should should be referenced

Reply: In Table 2, the references are given in the first column on the left.

(6) Figure 1 - I like the new figure, it is very helpful for the review but it looks like a rough sketch for a figure rather than a final publication quality figure. As this is central to the review, a higher quality version of the figure must be made. Perhaps information on the tissue of origin for OC, EC, and CC could also be included. Additional targets discussed in the review (ALPP, HER2) could also be shown in the diagram.

Reply: We have modified the figure as requested and improved its quality.

(7) New tables are very helpful. The formatting is a bit inconsistent, but I assume that would be fixed in the proof stage.

Reply: We were asked by some reviewers to integrate the tables with more information, hence their formatting has somewhat changed. Now we have reformatted the tables.

(8) Line 543 - Discussion of toxicity and the use of suicide genes is confusingly worded. This should be re-written to clearly explain why a suicide strategy addresses on-target off-tumor toxicity.

Reply: We have modified that sentence as follows (lines 534-536): “Thus, given the lack of specific antigens, suicide genes have been inserted into some types of CAR molecules, in order to induce selective apoptosis of CAR-T cells and avoid the risk of unmanageable toxicities”.

(9) Line 551 - Solid tumors are locating in specific tissue sites sides, and may or may not be well vascularized.

Reply: ok, we have modified as follows (line 542): “solid tumors are located in specific sites …”

Reviewer 3 Report

The manuscript has been extensively revised.

I don't have any new minor or major comments about the work.

Good job!!!

Author Response

The manuscript has been extensively revised. I don't have any new minor or major comments about the work. Good job!!!

Reply: We very much appreciate the reviewer's reply.

This manuscript is a resubmission of an earlier submission. The following is a list of the peer review reports and author responses from that submission.

Round 1

Reviewer 1 Report

This review covers the immune therapy history and rationale for gynecological tumors thoroughly. It starts with the known body of knowledge in the PD1/PD-L1. Then the paper moves to cover the structure and rationale for CAR-T utilization. 

This manuscript is very well written and easy to read. 

Few suggestions:

  • The text covering the previous experience with PD1/PDL1 treatments is long. I would suggest adding tables to summarize this section.
  • Adding a table to summarize the results from CAR-T therapy is helpful. 
  • CAR-T's are associated with prolonged lymphopenia, infections, cytokine release syndrome(CRS), and neurotoxicity (ICAN). This manuscript does not come across those complications. This paper is long. Therefore, I suggest adding a small paragraph in the discussion section to summarize the rates of CRS/ICAN that are so far reported. 

Author Response

Response to Reviewer 1 Comments:

This review covers the immune therapy history and rationale for gynecological tumors thoroughly. It starts with the known body of knowledge in the PD1/PD-L1. Then the paper moves to cover the structure and rationale for CAR-T utilization. 

This manuscript is very well written and easy to read. 

Few suggestions:

The text covering the previous experience with PD1/PDL1 treatments is long. I would suggest adding tables to summarize this section.

Reply: We thank the reviewer for his suggestions and have added a table (Table 1) to summarize previous experiences with PD1/PDL1 treatments.

Adding a table to summarize the results from CAR-T therapy is helpful.

Reply: We have created a new table (Table 2) that summarizes the main results of FDA-approved CAR-T cell therapy.

CAR-T's are associated with prolonged lymphopenia, infections, cytokine release syndrome(CRS), and neurotoxicity (ICAN). This manuscript does not come across those complications. This paper is long. Therefore, I suggest adding a small paragraph in the discussion section to summarize the rates of CRS/ICAN that are so far reported.

Reply: We have modified the Discussion as requested (lines 565-568).

Reviewer 2 Report

In the review mansuscript "Chimeric Antigen Receptor (CAR)-T cell therapy and its development in gynaecological (ovarian, endometrial and cervical) tumors", Schepisi et al have reviewed ongoing clinical trials of CAR-T treatment for gynecological tumor types, as well as presenting some relevant background on the molecular rationale and clinical development of immunotherapy for these hard to treat and high mortality tumors. 

Authors have presented a well organized and quite comprehensive review of the literature in this area, and in particular a review of the surprising number of clinical trials currently investigating CAR-T for gynecological cancers seems timely and useful. 

While the review does well to assemble the facts, it struggles to provide adequate context and discussion in some sections. In addition there are several grammatical and spelling errors throughout the document. For this reason, it is the opinion of this reviewer that the manuscript would require significant revision before being acceptable for publication.

Reworking the document to better focus and discuss the topic of clinical CAR-T development for gynecological cancers would be required. 

Major concerns:

(1) Additional context is required to synthesize the facts that are provided throughout the document. For example, in section 2.1 authors introduce the idea that tumor infiltrating lymphocytes (TILs) are present in 50% of OC, but authors fail to discuss what the implications of this fact are in regard to immunotherapy. Does the presence of TILs predict responsiveness to immunotherapy? What would the implication be for CAR-T therapy in particular? Does the presence of TILs correlate with CAR-T response?

There are many more examples of sections throughout the review where authors present a string of facts about various cancer types but fail to adequately discuss their relevance. See more specific examples in specific pointe below. 

(2) The manuscript would probable be improved by more specifically focusing on CAR-T rather than immunotherapy in general. 

(3) There are important factual errors in the introduction to CAR-T therapy. CAR-T was not "initially conceived as a treatment for hematological neoplasms". CAR-T was conceived as a hypothetical receptor that could be used to re-target T cell responses against surface antigens. Early data and even clinical trials were performed targeting HIV viral antigen (CD4-CAR) and CEA antigen. Authors should review the pre-history of CAR-T which occurred before successes were realized in hematological cancers (eg https://doi.org/10.3390/antib8030041)

It would be correct to state that "Clinical successes for CAR-T were first realized in hematological neoplasms..." but there has been a long development path for these successes. 

(4) Figure 1 recapitulates the typical structure of CARs that can be found in countless reviews. It seems of little value here, and unless there is specific relevance of different generation of CARs to gynecological cancers it should be removed. A figure providing an overview of the most relevant aspects of gynecological cancers with respect to CAR-T therapies would be much more useful. 

(5) The tables listing ongoing and completed clinical trials for CAR-T in gynecologcial cancers are very useful. It would be helpful to include a column with information regarding the status of each trial and a summary of results if they are available, as well as a reference to any published studies. 

(6) The discussion section is very short and does not add significantly to the review. If more discussion is provided throughout the review, this section be merged with the conclusions. 

Specific points:

Line 43 - "fourth diagnosed" should read "fourth most-commonly diagnosed"

Line 98 - "role pro-tumorigenic" should read "pro-tumorigenic role"

Line 99 - What is the specific relevance of TAMs to tumor immunotherapy?

Line 112 - Grammar

Line 113 - Grammar

Line 121 - Section fails to summarize a rationale for immunotherapy in EC

Section 3.1 - How do the ORR and PFS rates compare to what would be expected for standard of care?

Line 183 - What is the prevalence of MMR-deficiency in gynecological cancers? 

Line 196 - The word "interesting" does not seem right

Line 209 - No context provided for durvalumab results

Line 274 - Antigen binding domain could be scFv, or possible other entities like a nanobody or receptor binding domain

Line 275 - The hinge/spacer domain is also important in CAR function

Line 297 - See above

Tables 1 and 2 - Provide status/results/reference columns

Line 415 - Exponent format is incorrect

Author Response

Response to Reviewer 2 Comments

In the review manuscript "Chimeric Antigen Receptor (CAR)-T cell therapy and its development in gynaecological (ovarian, endometrial and cervical) tumors", Schepisi et al have reviewed ongoing clinical trials of CAR-T treatment for gynecological tumor types, as well as presenting some relevant background on the molecular rationale and clinical development of immunotherapy for these hard to treat and high mortality tumors. 

Authors have presented a well organized and quite comprehensive review of the literature in this area, and in particular a review of the surprising number of clinical trials currently investigating CAR-T for gynecological cancers seems timely and useful. 

While the review does well to assemble the facts, it struggles to provide adequate context and discussion in some sections.

Reply:  We have substantially modified the manuscript to improve context and discussion.

In addition there are several grammatical and spelling errors throughout the document. For this reason, it is the opinion of this reviewer that the manuscript would require significant revision before being acceptable for publication.

Reply: The manuscript has been revised by an native English speaker to correct grammatical/spelling errors and improve readability.

Reworking the document to better focus and discuss the topic of clinical CAR-T development for gynecological cancers would be required.

Reply:  We have modified the manuscript as requested.

Major concerns:

1) Additional context is required to synthesize the facts that are provided throughout the document. For example, in section 2.1 authors introduce the idea that tumor infiltrating lymphocytes (TILs) are present in 50% of OC, but authors fail to discuss what the implications of this fact are in regard to immunotherapy. Does the presence of TILs predict responsiveness to immunotherapy? What would the implication be for CAR-T therapy in particular? Does the presence of TILs correlate with CAR-T response?

RE: We have modified the text in accordance with the reviewer’s requests (lines 88-94).

There are many more examples of sections throughout the review where authors present a string of facts about various cancer types but fail to adequately discuss their relevance. See more specific examples in specific pointe below.

Reply: The specific modifications to the text are explained below in each point.

(2) The manuscript would probably be improved by more specifically focusing on CAR-T rather than immunotherapy in general.

Reply: We have substantially modified the manuscript in accordance with the reviewer’s comment.

(3) There are important factual errors in the introduction to CAR-T therapy. CAR-T was not "initially conceived as a treatment for hematological neoplasms". CAR-T was conceived as a hypothetical receptor that could be used to re-target T cell responses against surface antigens. Early data and even clinical trials were performed targeting HIV viral antigen (CD4-CAR) and CEA antigen. Authors should review the pre-history of CAR-T which occurred before successes were realized in hematological cancers (eg https://doi.org/10.3390/antib8030041) It would be correct to state that "Clinical successes for CAR-T were first realized in hematological neoplasms..." but there has been a long development path for these successes.

Reply:  We thank the reviewer for the suggestions and have modified the section as recommended.

(4) Figure 1 recapitulates the typical structure of CARs that can be found in countless reviews. It seems of little value here, and unless there is specific relevance of different generation of CARs to gynecological cancers it should be removed. A figure providing an overview of the most relevant aspects of gynecological cancers with respect to CAR-T therapies would be much more useful. 

Reply: We have created a Figure 1 that provides an overview of the most relevant aspects of gynecological cancers with respect to CAR-T therapies.

(5) The tables listing ongoing and completed clinical trials for CAR-T in gynecological cancers are very useful. It would be helpful to include a column with information regarding the status of each trial and a summary of results if they are available, as well as a reference to any published studies.

Reply:  As requested, in Table 3 and 4 we have added a column with information about the status of trials. However, the majority of these trials are still in the recruitment phase and consequently there are still no publications.

(6) The discussion section is very short and does not add significantly to the review. If more discussion is provided throughout the review, this section be merged with the conclusions.

Reply: We have improved the Discussion section.

Specific points:

Line 43 - "fourth diagnosed" should read "fourth most-commonly diagnosed" Reply:  This has been modified as requested (line 53).

Line 98 - "role pro-tumorigenic" should read "pro-tumorigenic role".

Reply: That sentence has been removed (line 113-15).

Line 99 - What is the specific relevance of TAMs to tumor immunotherapy?               Reply: This sentence has been removed (line 113-15).

Line 112 – Grammar

Reply: The manuscript has been thoroughly revised by a native English speaker.

Line 113 – Grammar.

Reply: The manuscript has been thoroughly revised by a native English speaker.

Line 121 - Section fails to summarize a rationale for immunotherapy in EC.

Reply: We have modified the section accordingly (lines 133-143).

Section 3.1 - How do the ORR and PFS rates compare to what would be expected for standard of care?

Reply: We have clarified this in the text (lines 171-172).

Line 183 - What is the prevalence of MMR-deficiency in gynecological cancers?

Reply: This information has been added to the text (lines 133-135).

Line 196 - The word "interesting" does not seem right.

Reply:  The word has been deleted (line 220)

Line 209 - No context provided for durvalumab results

Reply:  We have modified the sentence to make it clearer (lines 230-232).

Line 274 - Antigen binding domain could be scFv, or possible other entities like a nanobody or receptor binding domain.

Reply: We have modified the text to include this information (lines 314-332).

Line 275 - The hinge/spacer domain is also important in CAR function.

Re: We have commented on this in the text (lines 322-327).

Line 297 - See above

Reply: As above (lines 346-353).

Tables 1 and 2 - Provide status/results/reference columns

Reply: We have added a column with information on trial status (The ‘old’ Table 1 and 2 are now Table 3 and 4 in the revised manuscript), but  these studies are ongoing and as yet there are no related publications.

Line 415 - Exponent format is incorrect.

Reply: This has been corrected (line 452-453).

Reviewer 3 Report

The proposed title for the review is not really suitable as CAR T concern 1/3 of what is written about. A more appropriate title will be: “Immunotherapy and its development for gynecological cancers”.

The manuscript should be proofread by a native or a competent English writer. The review is poorly written and this doesn’t help to focus on the actual content. Some examples below:

Line 22,27 and 42 are similar or identical sentence!!!: “Gynecological tumors represent neoplasms with both high morbidity and mortality”. The authors should come with something different. Do you think it is really interesting to read three times the exact same things????

This is not an impossible task to avoid repetition. Moreover, words are missing in this sentence and should be written like that (for example): “gynecological tumors represent a significant proportion of neoplasms with both high morbidity and mortality”

You have chosen to write in American English but you have used British variant in the text (example:  haematological: British to be replaced by hematological: American, or gynaecological to be replaced by gynecological). Please double check that everything in the manuscript is using American English.

“but it ranks the second place in both incidence and mortality” replaced by “but it ranks to the second place in both incidence and mortality”

«Ovarian Cancer (OC) represents the 1.6% of all tumor incidence and the 1.3% of all cancer deaths, respectively» replaced by “Ovarian Cancer (OC) represents 1.6% of all tumor incidence and 1.3% of all cancer deaths, respectively»

«As regards Endometrial Cancer (EC)» replaced by “As regards to Endometrial Cancer (EC)»

«As is well known, gynecological»   replaced by “As it is well known, gynecological”

Example of editing where everything you want to “say” is just “say”:

«In this context, the development of Chimeric Antigen Receptor (CAR) technology represents a further step forward that has led to extraordinary results in the field of haematological cancers.»

replaced by

«In this context, the development of Chimeric Antigen Receptor (CAR) technology, that has led to extraordinary results in the field of hematological cancers, represents a new opportunity for the treatment of gynecological cancers.”

This following sentence is not clear: «However, gynecological tumors, due to their correlation with immunological mechanisms, could represent a type of tumor with promising possibilities also in the context of CAR-T cell therapy.» What is the meaning of “their correlation with immunological mechanisms”?

«in fact, its main role is protecting endometrium against sexual infections and other pathogens, but also allowing embryo implantation.»

replaced by

«in fact, its main role is to protect endometrium against sexual infections and to allow embryo implantation.»

«singular cell types» to be replaced by «single-cell type»

Table 1 is not clear. Add the CAR T  target in the table 1.

An example of unfocused review:

CAR-T cell therapy in EC

To date, no trials are ongoing to evaluate chimeric antigen receptor (CAR-)T cell therapy against EC. Only one outdated trial tested lymphokine-activated killer (LAK) (i.e. IL-2-stimulated peripheral mononucleate cells) in a cohort of patients with several abdominal malignancies. Unfortunately, the single EC patient treated did not demonstrate any benefit from the LAK therapy [85].

What is the point of making a paragraph on something not done!!! LAK is a CAR T? No. A more interesting development would have been to discuss potential CAR T targets for EC in this paragraph.

The CAR T is not working for solid cancer because first: “the still provisional nature of the data obtained from the numerous studies”. Are you serious?

The CAR T is not working for solid cancer because “the still poor mastery of the method”. You should develop what are the actual limitation of the different CAR T designs

«The third reason, no less important, seems to be related to the action of CAR-T in solid tumors.» This comment is general and should be developed. Why CAR T are currently not working in solid tumor? What are the mechanisms involved? Considerable progress has been made in our understanding of CAR T cell failure in solid cancers!!! nothing is mentioned here.

you describe the 4th generation CAR T but we are already the 5th!!!

Author Response

Response to Reviewer 3 Comments

Extensive editing of English language and style required.

Reply:  The manuscript has been thoroughly revised by a native English speaker.

The proposed title for the review is not really suitable as CAR T concern 1/3 of what is written about. A more appropriate title will be: “Immunotherapy and its development for gynecological cancers”.

Reply: We thank the reviewer for the suggestion and have modified the title as follows: “Immunotherapy and its development for gynaecological (ovarian, endometrial and cervical) tumors: from immune checkpoint inhibitors to chimeric antigen receptor (CAR)-T cell therapy”.

The manuscript should be proofread by a native or a competent English writer. The review is poorly written and this doesn’t help to focus on the actual content. Some examples below:

Reply:  The manuscript has been thoroughly revised by a native English speaker.

Line 22,27 and 42 are similar or identical sentence!!!: “Gynecological tumors represent neoplasms with both high morbidity and mortality”. The authors should come with something different. Do you think it is really interesting to read three times the exact same things???? This is not an impossible task to avoid repetition. Moreover, words are missing in this sentence and should be written like that (for example): “gynecological tumors represent a significant proportion of neoplasms with both high morbidity and mortality”.

Reply: We have had the manuscript revised by a native English speaker to improve its readability.

You have chosen to write in American English but you have used British variant in the text (example:  haematological: British to be replaced by hematological: American, or gynaecological to be replaced by gynecological). Please double check that everything in the manuscript is using American English.

Reply: This has been done.

 “but it ranks the second place in both incidence and mortality” replaced by “but it ranks to the second place in both incidence and mortality”

Reply: This sentence in question has been modified accordingly.  

«Ovarian Cancer (OC) represents the 1.6% of all tumor incidence and the 1.3% of all cancer deaths, respectively» replaced by “Ovarian Cancer (OC) represents 1.6% of all tumor incidence and 1.3% of all cancer deaths, respectively»

Reply: This has been corrected.

«As regards Endometrial Cancer (EC)» replaced by “As regards to Endometrial Cancer (EC)»  

Reply: This has been corrected.

«As is well known, gynecological»  replaced by “As it is well known, gynecological”

Reply: The native English speaker who reviewed the manuscript  confirmed that ‘As is well known…’ is perfectly correct.

Example of editing where everything you want to “say” is just “say”: «In this context, the development of Chimeric Antigen Receptor (CAR) technology represents a further step forward that has led to extraordinary results in the field of haematological cancers.» replaced by «In this context, the development of Chimeric Antigen Receptor (CAR) technology, that has led to extraordinary results in the field of hematological cancers, represents a new opportunity for the treatment of gynecological cancers.”

Reply: The sentence in question has been modified (lines 70-72).

This following sentence is not clear: «However, gynecological tumors, due to their correlation with immunological mechanisms, could represent a type of tumor with promising possibilities also in the context of CAR-T cell therapy.» What is the meaning of “their correlation with immunological mechanisms”?

Reply: We have removed the sentence.

«in fact, its main role is protecting endometrium against sexual infections and other pathogens, but also allowing embryo implantation.» replaced by «in fact, its main role is to protect endometrium against sexual infections and to allow embryo implantation.»

Reply: This sentence has been modified for clarity (lines 96-98).

 «singular cell types» to be replaced by «single-cell type»

Reply: This has been done (line 105). 

Table 1 is not clear. Add the CAR T  target in the table 1.

Reply: This has been done. The ‘old’ Table 1 is now Table 2 in the revised manuscript.

An example of unfocused review: CAR-T cell therapy in EC. To date, no trials are ongoing to evaluate chimeric antigen receptor (CAR-)T cell therapy against EC. Only one outdated trial tested lymphokine-activated killer (LAK) (i.e. IL-2-stimulated peripheral mononucleate cells) in a cohort of patients with several abdominal malignancies. Unfortunately, the single EC patient treated did not demonstrate any benefit from the LAK therapy [85]. What is the point of making a paragraph on something not done!!! LAK is a CAR T? No. A more interesting development would have been to discuss potential CAR T targets for EC in this paragraph.

Reply: This is a pertinent point. On the basis of the reviewer’s comment, we have completely revised this part of the text (lines 483-526).

The CAR T is not working for solid cancer because first: “the still provisional nature of the data obtained from the numerous studies”. Are you serious?

Reply: The sentence in question was referring only to gynecological tumors. However, to avoid further confusion, we have removed it from the text.

The CAR T is not working for solid cancer because “the still poor mastery of the method”. You should develop what are the actual limitation of the different CAR T designs.

Reply: As requested, we have provided more information on the limitations of the different CART-T designs (lines 557-564).

«The third reason, no less important, seems to be related to the action of CAR-T in solid tumors.» This comment is general and should be developed. Why CAR T are currently not working in solid tumor? What are the mechanisms involved? Considerable progress has been made in our understanding of CAR T cell failure in solid cancers!!! nothing is mentioned here.

Reply:  We have commented on these issues in the text (lines 569-585)

you describe the 4th generation CAR T but we are already the 5th!!!

Reply: We have added a specific comment on 5th generation CART-T cells

(lines 346-363).